# Molecular Dissection of TDP-43 as a Leading Cause of ALS/FTLD

**DOI:** 10.3390/ijms232012508

**Published:** 2022-10-19

**Authors:** Yoshitaka Tamaki, Makoto Urushitani

**Affiliations:** 1Department of Neurology, Graduate School of Medicine, Kyoto University, Kyoto 606-8507, Japan; 2Department of Neurology, Shiga University of Medical Science, Otsu 520-2192, Japan

**Keywords:** TDP-43, RNA metabolism, aggregation, phase separation, neurodegenerative diseases, ALS, FTLD

## Abstract

TAR DNA binding protein 43 (TDP-43) is a DNA/RNA binding protein involved in pivotal cellular functions, especially in RNA metabolism. Hyperphosphorylated and ubiquitinated TDP-43-positive neuronal cytoplasmic inclusions are identified in the brain and spinal cord in most cases of amyotrophic lateral sclerosis (ALS) and a substantial proportion of frontotemporal lobar degeneration (FTLD) cases. TDP-43 dysfunctions and cytoplasmic aggregation seem to be the central pathogenicity in ALS and FTLD. Therefore, unraveling both the physiological and pathological mechanisms of TDP-43 may enable the exploration of novel therapeutic strategies. This review highlights the current understanding of TDP-43 biology and pathology, describing the cellular processes involved in the pathogeneses of ALS and FTLD, such as post-translational modifications, RNA metabolism, liquid–liquid phase separation, proteolysis, and the potential prion-like propagation propensity of the TDP-43 inclusions.

## 1. Introduction

Amyotrophic lateral sclerosis (ALS) is the most common motor neuron disease characterized by progressive muscle weakness due to loss of upper and lower motor neurons, leading to death from respiratory failure, usually within four years [1]. Frontotemporal lobar degeneration (FTLD) is the second-most prevalent cause of neurodegenerative dementia after Alzheimer’s disease, characterized by progressive deficits in behavior, executive function, or language owing to the gradual loss of neurons in the frontal and temporal lobes of the brain [2]. Of note, 15% of ALS cases develop cognitive and behavioral impairment consistent with a typical definition of FTLD, and 12.5% of FTLD patients have concomitant motor neuron disease [3,4]. In 2006, positive inclusions of hyperphosphorylated and ubiquitinated TAR DNA-binding protein 43 (TDP-43) were identified as a pathological hallmark in ALS and a subgroup of FTLD [5,6]. Strikingly, TDP-43-positive inclusions are detected in the vast majority of all ALS cases (~97%) and ~45% of all FTLD cases (termed FTLD-TDP) [7]. The subsequent discoveries of mutations in the *TARDBP* gene encoding TDP-43 in both ALS cases and rare patients with FTLD have demonstrated that TDP-43 is fundamentally involved in the pathogenesis of ALS and FTLD-TDP [8,9,10,11]. Therefore, ALS and FTLD are now recognized as forming a continuum of broad neurodegenerative diseases presenting clinical, neuropathological, and genetic similarities. In addition, it has become increasingly clear over the past decade that several other neurodegenerative diseases such as limbic-predominant age-related TDP-43 encephalopathy (LATE) and Perry’s syndrome share a common pathological feature characterized by the presence of aberrant phosphorylation, ubiquitination, cleavage, and/or nuclear depletion of TDP-43 in neurons and glial cells, known as ‘TDP-43 proteinopathies’ [12,13,14].

TDP-43 was initially identified as a repressor protein associated with HIV-1 transcription, and it is a highly conserved and ubiquitously expressed RNA/DNA-binding protein belonging to the heterogeneous nuclear ribonucleoprotein (hnRNP) family [15]. TDP-43 has been shown to play a crucial role in multiple cellular homeostasis, including the regulation of mRNAs involved in neuronal development [16,17]. Although TDP-43 is strongly implicated in the pathogenesis of ALS and FTLD-TDP [18], the mechanism by which TDP-43 dysfunction or aggregation occurs and causes neurodegeneration remains unknown. It should be noted that TDP-43 proteinopathy is mediated by both the loss-of-function and the gain-of-toxic-function of TDP-43 proteins in complicated manners [19,20]. Thus, a further understanding of the biological functions of TDP-43 and its pathological mechanisms has important implications for developing potential therapeutic strategies. This review article focuses on the physiological functions and multiple pathways involved in the disease mechanisms of ALS and FTLD-TDP.

## 2. Physiological Structure and Functions of TDP-43

TDP-43 contains 414 amino acids (aa) and is encoded by the *TARDBP* gene on chromosome 1 (1.p36.22). The structural domains comprise the following: an N-terminal region (aa 1–102) with a nuclear localization signal (NLS, aa 82–98); two RNA recognition motifs: RRM1 (aa 104–176) and RRM2 (aa 192–262); a nuclear export signal (NES, aa 239–250), a C-terminal region (aa 274–414) harboring a prion-like glutamine-/asparagine-rich (Q/N) domain (aa 345–366) and a glycine-rich region (aa 366–414) (Figure 1) [21,22,23,24,25,26].

Under physiological conditions, TDP-43 is natively dimeric or exists in a monomer-dimer equilibrium [27]. The N-terminal region regulates TDP-43 physiological monomer-homodimerization transition and RNA splicing and mediates aggregate formation [28]. This region promotes the self-oligomerization of TDP-43 in a concentration-dependent manner and enhances its DNA-binding affinity [29]. Moreover, the N-terminal domain-driven dimerization has been implicated mainly in RNA splicing [30], and this region is also involved in liquid–liquid phase separation (LLPS) [31]. By crystal analysis, the RRM2 domain is shown to form a homodimer upon interacting with DNA, mediated by the intermolecular hydrogen bonds of Glu246-Ile249 and Asp247-Asp247 [32]. Considering that the nuclear exclusion of TDP-43 may lessen DNA interaction, Glu246 and Asp247 could serve as early epitopes of mislocalized TDP-43 and may well be molecular targets of ALS [33,34].

RRM1 and RRM2 domains in TDP-43 are indispensable for RNA/DNA binding to regulate the transcription, translation, splicing, and stability of mRNA [35]. These domains bind to RNA/DNA with higher specificity toward UG/TG-rich sequences [23,24]. TDP-43 regulates the RNA splicing of non-conserved cryptic exons, maintaining intron integrity [36], and it also self-regulates the amount via TDP-43 splicing [37]. TDP-43 forms cytoplasmic RNP granules that undergo bidirectional microtubule-dependent transport in neurons and facilitate the delivery of mRNA to distal neuronal compartments [38]. Structure-function analysis has revealed that the two cysteine residues (Cys-173 and Cys-175) in the RRM1 domain are crucial for TDP-43′s conformation, and the substitution of these residues into serine results in the acceleration of amyloid fibrils of RRM1 and the disulfide-independent aggregate formation of TDP-43 [39]. The RRM2 domain may contribute to the dimerization of the TDP-43 protein [32]. The two cysteine residues in the RRM2 domain (Cys-198 and Cys-244) are associated with the formation of disulfide-bonded dimers, subsequently assembling into aggregate particles under oxidating conditions [40].

The C-terminal region of TDP-43, which harbors most of the ALS-associated *TARDBP* gene mutations and phosphorylation sites, strongly contributes to the pathological behavior of TDP-43 and is intrinsically disordered and aggregation-prone [41]. This region is physiologically involved in LLPS and forms membraneless dynamic protein droplets. LLPS may underlie stress granule formation, which protects the neuronal cells against cellular insults such as oxidative stress [42].

As a DNA/RNA binding protein, TDP-43 is involved in multiple aspects of RNA metabolism, including splicing, microRNA (miRNA) biogenesis, RNA transport and translation, and stress granule formation by interacting with numerous hnRNPs, splicing factors, and microprocessor proteins [7]. Given that TDP-43 performs such essential functions in cellular homeostasis, it is unsurprising that the structural and functional damages of TDP-43 may easily cause severe diseases such as ALS.

## 3. Pathogenic Mechanisms of TDP-43

In contrast to the relatively stereotyped pathology of TDP-43 aggregates in the vast majority of sporadic ALS and FTLD-TDP, the underlying mechanisms are varied and remain uncovered (Figure 2).

### 3.1. Mutant TDP-43 Proteins

Numerous pathogenic mutations in the TARDBP gene (>50 mutations) have been identified in both sporadic and familial cases of ALS and FTLD-TDP, accounting for six percent of global familial ALS (FALS) patients. The mutations are exclusively found in the C-terminal glycine-rich region [11,15]. Mutations in this particular region enhance the intrinsic aggregation propensity and cytotoxicity of the TDP-43 protein—as shown in recombinantly expressed TDP-43 protein harboring FALS-linked mutations, such as Q331K, M337V, Q343R, N345K, R361S, and N390D in vitro—and promote cytotoxicity in the yeast cells [43]. Aggregated TDP-43 peptides with A315T mutation cross-seed other TDP-43 peptides and an amyloid-β peptide [44]. A315T mutant TDP-43-derived amyloid fibrils cause the neuronal death of primary cultured neurons [45]. TARDBP mutations display TDP-43 proteinopathy and ALS symptoms identical to sporadic ALS. However, several TDP-43 mutations with higher aggregation propensity, such as G294V, A315T, M337V, G376D, and A382T, facilitate the cytoplasmic mislocalization of TDP-43 [46,47,48]. Moreover, TDP-43 mutations such as D169G, A315T, Q343R, G348C, and R361S result in the formation of larger stress granules and the reduction in their distribution density and mobility, thereby impairing RNA homeostasis and leading to neuronal dysfunction [41]. Longer half-lives of mutant TDP-43 proteins may also correlate with accelerated disease onset. For instance, ALS- and FTLD-associated TDP-43 mutations such as D169G, K263E, G298S, A315T, M337V, Q343R, G348C, N352S, and A382T are associated with the accelerated disease onset and enhanced TDP-43 aggregates in ALS [49,50]. In recent years, several reports have been published using TDP-43 knock-in mouse models. In homozygous TDP-43 M337V knock-in mice, mRNAs splicing of Kcinp2, Sort1, and Sema3f is deregulated in the spinal cords, although they develop normally without exhibiting detectable motor dysfunction and neurodegeneration [51]. TDP-43 Q331K knock-in mice demonstrate cognitive dysfunction and perturbed TDP-43 autoregulation, leading to the altered splicing of a pivotal dementia-associated gene, Mapt [52]. Heterozygous TDP-43 N390D knock-in mice show age-dependent motor dysfunction and muscle atrophy [53].

### 3.2. Post-Translational Modifications

Various post-translational modifications of TDP-43, including cleavage, ubiquitination, and phosphorylation, have been implicated in neurotoxicity in the TDP-43 proteinopathies. The aberrant mislocalization of TDP-43 triggers various post-translational modifications, although the precise mechanisms remain elusive [41].

#### 3.2.1. Cleavage

The generation of 25–35 kDa C-terminal fragments (CTFs) of TDP-43 through proteolytic cleavages by the caspase and calpain proteases is reported as one of the prominent pathological processes in ALS and FTLD-TDP. The C-terminal region of TDP-43 harbors most of the ALS-associated mutations and phosphorylated sites, and it is intrinsically disordered and aggregation-prone [54]. The CTFs of TDP-43 are the major components of the inclusions in the ALS-affected tissues and are highly cytotoxic [55]. The C-terminal region of TDP-43 also contains a short, highly dynamic, and unstable helix-turn-helix region (aa 311–360) which can efficiently form amyloid-like fibrils [22]. In this regard, the crucial roles of N-terminal fragments have also been reported [56].

#### 3.2.2. Ubiquitination

Ubiquitinated TDP-43 inclusions are a pathological hallmark in the ALS- and FTLD-TDP-affected brain [5,6]. The E3 ubiquitin ligase ubiquitinates TDP-43 proteins via the ubiquitin lysines, Lys-48 and Lys-63, which promote the cytoplasmic accumulation of TDP-43 through a multiprotein complex with histone deacetylase 6 (HDAC6) [57]. The full-length TDP-43 aggregates are labeled by both Lys-48- and Lys-63-linked polyubiquitin chains and subsequently directed toward different proteolysis pathways. The Lys-48-linked polyubiquitin chains induce ubiquitin proteasomal-mediated proteolysis of TDP-43, while the Lys-63-linked polyubiquitin chains cause autophagy-mediated clearance [58].

#### 3.2.3. Phosphorylation

In addition to ubiquitination, hyperphosphorylated TDP-43 is a pathological feature in ALS and FTLD-TDP [5,6]. Whether phosphorylation occurs at the early or late phase of TDP-43 proteinopathy is not conclusively known. The phosphorylation of TDP-43 triggers cytoplasmic mislocalization and accumulation of TDP-43 in cultured neuronal cells [59]. On the other hand, mislocalized TDP-43 is not necessarily phosphorylated in transgenic mice, suggesting that the phosphorylation of TDP-43 is a secondary phenomenon after TDP-43 misfolding. TDP-43 has multiple phosphorylation sites in carboxyl-terminal regions. Although the physiological function of TDP-43 phosphorylation is controversial, a recent report suggests that TDP-43 hyperphosphorylation suppresses TDP-43 aggregation and confers a protective cellular effect [60]. Phosphorylation at Ser-379, Ser-403, Ser-404, and especially Ser-409/Ser-410 is mediated by kinases—such as casein kinase-1 and -2 (CK1 and CK2), cell division cycle 7 (CDC7), and tau tubulin kinase-1 and -2 (TTBK1 and TTBK2)—which leads to increased oligomerization and fibrillization of TDP-43 and is now considered a signature of TDP-43 pathology in ALS and FTLD-TDP [61]. The pathologically phosphorylated TDP-43 forms a protease-resistant fibril structure, which is also a significant biological characteristic of TDP-43 proteinopathies [62].

#### 3.2.4. SUMOylation

The covalent attachment of small ubiquitin-like modifier (SUMO) proteins to specific proteins, termed SUMOylation, is a reversible pathway that competes with ubiquitin to alter subcellular localization and protein turnover. SUMOylation regulates the functional properties of the specific proteins in the nucleus and cytoplasm of neurons, playing a role in the cellular responses to hypoxia, oxidative stress, glutamate excitotoxicity, and proteasome impairment, pivotal processes linked to motor neuron degeneration in ALS [63]. The mutation of the unique SUMOylation site at Lys-136 of TDP-43 (K136R), designed as a SUMOylation-resistant TDP-43 plasmid, has a less effective exon-skipping activity than the wild-type (WT) in HEK293T cells. Moreover, the K136R mutant TDP-43 inhibits the mislocalization of the WT upon promoting de-SUMOylation, while this mutant TDP-43 is recruited into the stress granules to a lower extent than the WT in SK-N-BE cells exposed to sodium arsenite [64]. Therefore, SUMOylation contributes to the TDP-43 subcellular localization and recruitment to stress granules after oxidative stress, in addition to the modification of TDP-43 splicing activity.

#### 3.2.5. Acetylation

Lysine acetylation is a major covalent modification controlling diverse cellular processes and has been implicated in Alzheimer’s disease and other neurodegenerative diseases [65]. Among 20 lysine residues in the TDP-43 molecule, Lys-145 in RRM1 and Lys-192 in RRM2 are especially prone to acetylation. TDP-43 acetylation impairs RNA binding and mitochondrial functions and promotes the accumulation of insoluble and hyperphosphorylated TDP-43 aggregates. Oxidative stress facilitates the acetylation and aggregation of TDP-43 in cultured neuronal cells [66]. Notably, an antibody raised against the acetylation at the Lys-145 identifies acetylated TDP-43 in the ALS patient’s spinal cord, indicating that aberrant TDP-43 acetylation crucially underlies the loss of RNA binding in ALS/FTLD [66,67]. Recently, TDP-43 acetylation has been found to promote phase separation into intranuclear liquid spherical shells with liquid cores, termed anisosomes. These structures convert to aggregates when ATP levels are reduced, thus indicating that anisosomes may be the antecedents of pathological aggregates in TDP-43 proteinopathies [68].

### 3.3. Nuclear TDP-43 Depletion

TDP-43 functions as a DNA/RNA binding protein and is predominantly located in the nucleus of cells. It shuttles between the nucleus and the cytoplasm, engaging in diverse cellular functions within both compartments [15]. This nucleo-cytoplasmic shuttling is regulated by the nuclear localization signal (NLS) and the nuclear export signal (NES) and their relevant molecules, such as importins. Therefore, the irreversible mislocalization of TDP-43 may cause severe loss of function, leading to neurodegeneration [69].

TDP-43 interacts with several proteins involved in the mRNA metabolisms in the nucleus and the cytoplasm [7]. For this reason, TDP-43 continuously shuttles between the nucleus and cytoplasm in a transcription-dependent manner, and its cellular concentration is tightly auto-regulated to maintain its steady levels via a feedback loop [70]. Although the precise mechanisms of the pathological TDP-43 mislocalization remain elusive, nuclear TDP-43 depletion appears to precede the formation of TDP-43 inclusions [71]. TDP-43, devoid of NLS, is located in the cytoplasm, forms the aggregates, and sequesters the native TDP-43. This step promotes the reduction in nuclear TDP-43 in transgenic mice, which consequently alters the transcripts that regulate chromatin assembly and histone processing [72]. Moreover, nuclear TDP-43 reduction leads to RNA degradation and reduction, especially while encoding a protein involved in synaptic activity [16]. However, it should be noted that only the forced relocation of TDP-43 in the cytoplasm does not suffice to replicate typical TDP-43 pathologies such as phosphorylated inclusions [39,73].

### 3.4. Dysregulation of RNA Metabolism

TDP-43 is involved in all aspects of RNA metabolism ranging from splicing, transcription, transport, storage into RNA/protein granules, and translation. Accumulating evidence has demonstrated that dysregulation of RNA metabolism contributes to ALS pathogenesis [7,74]. With the advancement of new technologies, including RNA sequencing, the link between TDP-43 pathology and RNA homeostasis is being extensively uncovered. An optogenetic approach using a Cry2-TDP-43 construct has demonstrated that the formation of TDP-43 inclusions is driven by aberrant interactions between low-complexity domains of TDP-43 that are inhibited by RNA binding [75]. A crosslinking and immunoprecipitation (CLIP) analysis has shown that altered condensation properties of TDP-43 selectively modify its RNA-regulatory network [76].

TDP-43 has been shown to function as a splicing regulator and its nuclear depletion results in the mRNA splicing aberrations of multiple RNA targets [16]. In addition, the ALS-associated TDP-43 mutation (Q331K) alters mRNA splicing processes in transgenic mice. Mutant TDP-43 (Q331K) reduces splicing processes in target RNAs, such as *Kcnip2*, *Abhd14a*, *Ctnnd1*, and *Atp2b1*, demonstrating a loss of normal TDP-43 function. In contrast, the Q331K mutant enhances the splicing of other target RNAs, such as *Eif4h* and *Taf1b* [77].

The neuronal growth-associated factor, stathmin-2, is an attractive candidate for the splicing target of TDP-43. Stathmin-2 is necessary for normal axonal outgrowth and regeneration, and lowered TDP-43 levels reduce its binding to sites within the first intron of stathmin-2 pre-messenger RNA and produce a truncated and non-functional mRNA. Of note, reduced stathmin-2 expression is associated with poor axonal regeneration, which is rescued by normalization of stathmin-2 expression [78,79].

TDP-43 also regulates a cryptic exon UNC13A splicing event. TDP-43 depletion from the nucleus induces robust inclusions of a cryptic exon in UNC13A mRNA and reduced UNC13A protein expression [80,81].

The gain of splicing function has been recently demonstrated in knock-in mice carrying TDP-43 missense mutation: TDP-43 autoregulation is perturbed, leading to altered splicing of *Mapt* in TDP-43 Q331K knock-in mice [52]; TDP-43 M323K knock-in mice induce a novel gain of splicing activity leading to skipping of certain constitutive exons, albeit this mutation is not an ALS-linked mutation [82]; mRNAs splicing of *Kcinp2*, *Sort1*, and *Sema3f* is deregulated in the spinal cords of TDP-43 M337V knock-in mice [51].

The examination of splicing patterns of TDP-43 target genes in lower motor neurons of postmortem ALS cases has revealed the widespread dysregulations of mRNA splicing that specifically affected genes involved in ribonucleotide binding [83]. Moreover, TDP-43 regulates the splicing of non-conserved cryptic exons which are spliced into messenger RNAs in the absence of TDP-43, and the repression of cryptic exons is impaired in ALS-FTLD cases, suggesting that the splicing defect potentially underlies TDP-43 proteinopathy [36]. Together, these studies suggest that loss of function TDP-43 in RNA splicing defects is a common mechanism in ALS.

TDP-43 forms cytoplasmic mRNP granules that undergo bidirectional, microtubule-dependent transport in neurons and facilitate the delivery of target mRNA, such as neurofilament light (NEFL) mRNA, to distal axonal compartments; notably, TDP-43 mutations (A315T and M337V) impair this mRNA transport function [38]. Likewise, TDP-43 also binds G-quadruplex-forming mRNAs and transports them to distal neurites. The disease-associated mutation in TDP-43 (M337V) lacks the activity of binding and transport of G-quadruplex-containing mRNAs [84]. TDP-43 within neurons regulates the transport of specific mRNA targets in mRNP granules along axonal microtubules to distal neurites.

### 3.5. Nucleocytoplasmic Transport Dysfunction

While TDP-43 is predominantly located in the nucleus, it continuously shuttles between the nucleus and cytoplasm in a transcription-dependent manner, engaging in diverse physiological cellular functions [70]. TDP-43 binds to a nuclear import factor, karyopherin-α, thereby confirming the nuclear import pathway for the import of TDP-43 [85]. A nuclear import receptor, karyopherin-β1, is a part of the nuclear pore machinery harboring chaperone effects, and it decreases TDP-43 fibrillization possibly by associating with its nuclear localization signal sequence [86]. TDP-43 aggregation in the cytoplasm, but not in the nucleus, reportedly induces the sequestration and mislocalization of various functional proteins. Thus, nucleocytoplasmic transport dysfunction may contribute to the cellular pathology of aggregate deposition diseases such as ALS and FTLD [87]. For instance, aggregated and disease-linked mutant TDP-43s (Q331K and M337V) trigger the sequestration and/or mislocalization of nucleoporins and transport factors, leading to interference with nuclear protein import and RNA export. Moreover, nuclear pore pathology is present in brain tissue in cases of ALS and those involving genetic mutations in *TARDBP* genes, implicating TDP-43-mediated nucleocytoplasmic transport defects as a common disease mechanism in ALS and FTLD-TDP [88].

### 3.6. Stress Granules

Eukaryotic cells have developed mechanisms that protect cells against exposure to multiple cellular stresses such as heat shock, oxidative stress, hyperosmolarity, viral infection, and chemical exposure. Non-membranous cytoplasmic foci with a size range of 0.1–2.0 μm are formed as stress granules (SG) in response to these diverse environmental conditions [89,90]. The SGs consist of RNA-binding proteins, polyA-binding proteins, 40S ribosomal subunits, protecting mRNAs and eukaryotic initiation factors [91]. The SGs are formed reversibly, dissolving after the cellular stress is over [92]. However, failure of the stress response may facilitate the conversion of SGs into pathological inclusions observed in ALS and FTLD [93,94,95,96]. On the other hand, recent studies have reported that pathological TDP-43 inclusions are formed under stress conditions independently of SGs: increased concentration of TDP-43 in the cytoplasm provokes long-lived liquid droplets of cytosolic TDP-43 whose assembly and maintenance are independent of SGs [97]; transient oxidative stress, proteasome inhibition or inhibition of the ATP-dependent chaperone activity of HSP70 provokes reversible cytoplasmic TDP-43 de-mixing and transition from liquid to gel/solid, independently of RNA binding or SGs [98]. TDP-43 contributes to both the assembly and maintenance of SGs under oxidative stress, containing T-cell intracellular antigen 1 (TIA-1) and RasGAP-association endoribonuclease (G3BP), essential component proteins for SGs [99]. The ALS-linked TDP-43 mutations are strongly associated with SG dynamics. For instance, the ALS-linked D169G and R361S mutant TDP-43 proteins are more resistant to SG disassembly than the WT in patient lymphoblast cells [99]. Another ALS-linked G348C mutant forms larger SGs and is more readily incorporated into the SGs than the WT TDP-43 [100]. In addition, the ALS-linked A315T and Q343R mutants increase the sizes of TDP-43-containing RNA granules and decrease their distribution density and mobility in the dendritic arbor of rat hippocampal neurons [101]. Based on the fact that NLS-defective TDP-43 proteins prevent SG formation [102], it is suggested that an excess amount of cytosolic TDP-43 may impair SG formation and that TDP-43 mislocalization may crucially affect SG dynamics.

### 3.7. Liquid–Liquid Phase Separation

Liquid–liquid phase separation (LLPS) is a process that mediates the formation of membraneless, spherical, liquid droplet-like organelles, in which proteins containing prion-like domains are involved. This process is a vital pathway in various neurodegenerative diseases [15]. The low-complexity C-terminal domain of TDP-43 is responsible for the LLPS of SGs and cytoplasmic bodies observed in ALS and FTLD-TDP [100]. In addition, dimerization and oligomerization of the N-terminal domain are also responsible for LLPS [31]. Under stress conditions such as oxidative and osmotic stress, TDP-43 protein is incorporated into SGs, whereas LLPS may convert to cytoplasmic aggregates of TDP-43 in the pathological state, including chronic stresses [75]. Under physiological conditions, TDP-43 is localized to the nuclear membraneless organelles, such as the Gemini of Cajal bodies (Gems), promyelocytic leukemia (PML) nuclear bodies, or paraspeckles [103]. In ALS, aberrant accumulation of TDP-43 impairs the formation of Gems, leading to defective spliceosome function [104,105]. While LLPS is reversible in WT TDP-43, several mutations in the C-terminal region, such as A321G, Q331K, and M337V, affect the transitional state of the LLPS, increasing the propensity to aggregate [42]. Structural analysis has shown that a tryptophan residue, W334, in the a-helical segment (aa: 320–340) is crucial for the phase separation of TDP-43 [106,107]. Moreover, a malfunctioning LLPS process inhibits nucleocytoplasmic transport and induces the aberrant mislocalization of nuclear TDP-43. The poly(ADP-ribose) polymerase called tankyrase modifies TDP-43 by adding a poly(ADP-ribose) polymer to its NLS residues and promotes LLPS and TDP-43 aggregation [108]. However, existing evidence is not sufficient to conclude that TDP-43 aggregates generated by LLPS can seed the further aggregation of TDP-43.

### 3.8. Oligomerization

The oligomer formation of TDP-43 is an intermediary step that occurs during the formation of large aggregates. The TDP-43 oligomerization stage is initiated by its RNA-binding region. Specific binding to GU-rich RNA strongly intercepts TDP-43 oligomerization and the resultant aggregate formation, suggesting that RNA is a crucial regulator in maintaining TDP-43 solubility [109]. TDP-43 oligomers are stable and neurotoxic in vitro and in vivo, and interestingly, they function as cross-seeds for Alzheimer’s amyloid-β to form amyloid oligomers, demonstrating interconvertibility between the amyloid species. Indeed, such oligomers are present in the forebrains of transgenic TDP-43 mice and FTLD patients [110].

### 3.9. Mitochondrial Dysfunction

Mitochondrial dysfunction has been implicated in the mechanism of TDP-43 toxicity. Since post-mitotic neurons have high demands for mitochondria due to synaptic homeostasis, mitochondrial dysfunction in neurons significantly affects cellular function and survival [111,112]. Mitochondrial dysfunction has been demonstrated using in vivo and in vitro models which express the WT TDP-43 or its mutants. In the primary motor neurons, over-expression of WT TDP-43 or its ALS-associated mutants (Q331K and M337V) shortens the mitochondrial length and affects mitochondrial movement, which could be avoided by the co-expression of the mitochondrial fusion protein, mitofusin-2 (Mfn2) [113]. The transgenic mice expressing the mutant TDP-43 (A315T) demonstrate defective mitochondrial transport and morphology in their neurons [114]. Furthermore, mitochondrial fission and fragmentation are highly enhanced in the muscle and motor neurons of the TDP-43 *Drosophila* model, thereby suggesting an imbalance of the mitochondrial dynamics [115]. In the yeast model, the expression of TDP-43 triggers increased oxidative stress, apoptosis, necrosis, and the formation of peri-mitochondrial TDP-43 aggregates. Therefore, the functional mitochondria may exacerbate the deleterious effects of TDP-43, suggesting the roles of mitochondria as double-edged swords in TDP-43-linked neurodegeneration [116]. Fractions of FALS-linked TDP-43 mutants translocate to mitochondria, whereby the interaction of TDP-43 with mitochondria-transcribed mRNAs encoding respiratory complex I subunits is more robust than WT TDP-43 in physiological conditions. This aberrant interaction may underlie the impairment of their expression and the disassembly of complex I in TDP-43-ALS. Moreover, the inhibition of TDP-43 translocation to mitochondria abolishes mitochondrial dysfunction and neuronal loss and ameliorates the phenotypes of transgenic mutant TDP-43 mice in both WT and mutant TDP-43 [117]. Interestingly, mitochondrial caspases reportedly cleave TDP-43 into the pathological C-terminal fragments of 25 kDa and 35 kDa [118], although the molecular linkage between the carboxyl terminus of TDP-43 and mitochondrial dysfunction is unanswered. TDP-43 pathology is associated with a neuroinflammatory cytokine profile related to the upregulation of nuclear factor κB (NF-κB) and type I interferon (IFN) pathways. This inflammation is driven by the cytoplasmic DNA sensor cyclic guanosine monophosphate (GMP)-AMP synthase (cGAS) when TDP-43 invades mitochondria and releases mitochondrial DNA via the permeability transition pore [119].

### 3.10. Oxidative Stress

Reactive oxygen species (ROS) arise as by-products of aerobic metabolism. Most cellular ROS originate from the leaked electrons from the mitochondrial respiratory chain. The oxidative phosphorylation unavoidably produces ROS, such as hydrogen peroxide (H_2_O_2_) and the superoxide radical anion [120,121,122]. Cohen et al. demonstrated that oxidative stress promotes TDP-43 insolubility via the cross-linking of TDP-43, the amount of which is increased in ALS and FTLD brain samples [66]. Moreover, WT or mutant (Q331K or M337V) TDP-43 proteins reportedly increase ROS in the NSC34 cell line. Importantly, the ROS production is also accelerated by truncated TDP-43 (CTFs TDP-25 or TDP-35) [123]. Moreover, Guerrero et al. recently showed that in SH-SY5Y cells expressing mutant TDP-43 (p.Q331K), increased ROS causes DNA strand breaks and neuronal apoptosis [124].

### 3.11. ER Stress

Most secretory proteins in eukaryotes enter the endoplasmic reticulum (ER) after translation and are folded and assembled. The ER responds to the burden of unfolded proteins in its lumen (ER stress) by activating various intracellular signaling pathways, termed the unfolded protein response (UPR) [125]. Large amounts of calcium (Ca^2+^) are stored in the ER, crucially involved in the ER-mitochondrial calcium cycle, which may link mitochondrial energy production and ER protein processing with neuronal synaptic activity [126]. Inappropriately handled ER stress results are implicated in the pathogenesis of ALS and FTLD [126,127]. TDP-43 perturbs an interaction between the ER protein known as vesicle-associated membrane-protein-associated protein-B (VAPB) and the mitochondrial protein tyrosine phosphatase interacting protein 51 (PTPIP51), which together regulate the association between the ER and mitochondria. Overexpression of WT or ALS-associated mutant TDP-43 (M337V, Q331K, A382T, G348C) interrupts the ER-mitochondria interaction, while TDP-43 is also associated with the activation of glycogen synthase kinase-3 (GSK-3) and disturbs Ca^2+^ homeostasis [128]. Mutant TDP-43 (A382T or M337V) impairs and reduces Ca^2+^ signaling from the ER compared with WT TDP-43 in cell lines, suggesting that ER plays a crucial role in Ca^2+^ signal homeostasis in ALS [48]. In *C. elegans* (worm) and *D. rerio* (zebrafish) models expressing mutant TDP-43 (G348C), Vaccaro et al. showed that ER stress suppression with pharmacological compounds is neuroprotective against TDP-43 proteinopathy [129], and the ER stress is a promising therapeutic target with respect to therapeutic research for ALS and FTLD.

### 3.12. Impairment of Protein Quality Control (Ubiquitin and Autophagy Dysfunction)

There are two major proteolytic pathways in eukaryotic cells: the ubiquitin-proteasome system (UPS) and the autophagy-lysosome pathway, which control protein quality and maintain cellular homeostasis. The frequent presence of cytoplasmic aggregates of TDP-43 is tightly linked to proteostasis dysfunction in ALS [130,131,132]. Both full-length and cleaved TDP-43 are ubiquitinated and degraded via UPS or autophagy. Moreover, TDP-43 aggregates in ALS/FTLD contain carboxyl fragments and full-length species. Further, both the soluble and the aggregated TDP-43 are degraded by the UPS and autophagy [133,134,135]. Therefore, the role of UPS in discriminating the physiological TDP-43 and aberrant fragments remains unanswered.

In neurodegenerative diseases, the UPS overwhelming may result in inappropriate protein degradation and aberrant protein accumulation, which can lead to cellular apoptosis [136]. A recent report showed that SH-SY5Y cells stably expressing disease-associated TDP-43 mutant (G298S and A382T) proteins have an accelerated turnover compared with WT TDP-43, and the degradation of these mutant proteins is partially prevented by a proteasome inhibitor but not by a lysosomal inhibitor, suggesting that they are degraded by the UPS [137]. Inhibition of the UPS by a proteasome inhibitor also induces the phosphorylation, ubiquitination, and cytoplasmic aggregation of TDP-43 in cultured cells, recapitulating the major pathological features of TDP-43 proteinopathies [138]. Moreover, conditional knockout mice of the proteasome subunit Rpt3 in a motor neuron-specific manner (Rpt3-CKO) lead to TDP-43 mislocalization in motor neurons and develop an ALS phenotype accompanied by progressive motor neuron loss and gliosis [139]. This in vivo study suggests that the UPS may predominate the quality control regarding TDP-43 proteinopathies to autophagy.

TDP-43 functions as an autophagy regulator by associating with the mRNA for autophagy-related 7 (ATG7) and some of the ALS-linked TDP-43 mutations lose their ATG7 mRNA-binding ability [140]. TDP-43 also affects the localization of the transcription factor EB (TFEB) that controls the expressions of several autophagy lysosomal pathway proteins [141]. Moreover, cytoplasmic inclusions are frequently positive for autophagy markers such as LC3 and p62/SQSTM1 in the brain or spinal cord sections in ALS/FTLD [131,142]. It is suggested that an accumulation of autophagosomes during autophagy can overwhelm the autophagy-lysosome system in neurodegenerative diseases [143]. In ALS and FTLD-TDP, the loss of TDP-43 functions due to cytoplasmic mislocalization is implicated in autophagosome-lysosome fusion and autophagy-mediated protein degradation [141].

Ubiquilin (UBQLN), a member of the ubiquitin-like (UBL)-ubiquitin-associated (UBA) family, is a dual regulator of both the proteasomal and autophagic protein degradation system. Mutations in the *UBQLN2* gene encoding the ubiquitin-like protein ubiquilin 2 cause X-linked ALS/FTLD, and UBQLN2-positive inclusions are identified in ALS with *UBQLN2* mutations as well as in cases of both familial and sporadic ALS without *UBQLN2* mutations [144]. Mutations in the ALS-associated *UBQLN2* gene impair autophagic protein degradation and promote TDP-43 aggregation in neuronal cells [145]. Moreover, UBQLN2 proteins exacerbate TDP-43 pathology by competing with the UPS for binding to ubiquitin in double-transgenic UBQLN2 P497H; TDP-43 G348C mice [146].

### 3.13. Axonal Transport Impairment

Axonal transport is the process whereby motor proteins actively navigate microtubules to deliver cargoes, such as organelles, cytoskeletal elements, and growth factors, from one end of an axon to the other and is widely regarded as essential for nerve development, function, and survival [147,148]. Impairments in axonal transport have been demonstrated in several ALS models. Transgenic mice expressing mutant TDP-43 (M337V) display axonal transport perturbations preceding symptom onset, while mutant FUS show normal function [149]. TDP-43 is actively transported along motor neuron axons and colocalizes with axonal mRNA-binding proteins, such as FMRP, IMP1, HuD, and SMN, in motor neuron axons [150]. Overexpression of either full-length (WT) or mutant (M337V or A382T) TDP-43 in primary motor neurons prevents axon outgrowth, and mutant TDP-43 types are more abundant in the axons than the WT [150]. TDP-43 forms cytoplasmic mRNA granules in *Drosophila* motor neurons, primary murine cortical neurons, and stem-cell-derived motor neurons, which facilitate the delivery of target mRNA to distal neuronal compartments. ALS-associated TDP-43 mutants (M337V and A315T) impair the mRNA transport function [38]. Furthermore, mitochondrial axonal transport is observed in transgenic mice overexpressing either WT or ALS-associated mutant (A315T, Q331K or M337V) TDP-43 [113,114]. Therefore, it is conceivable that TDP-43 dysfunction in axonal transport is crucially involved in ALS/FTLD pathogenesis.

### 3.14. Prion-like Propagation

Accumulating evidence indicates that the spatial progression of neurodegenerative diseases is mediated by prion-like cell-to-cell propagation of disease-related proteins, in which the template-induced misfolding of normal endogenous proteins to pathological conformations occurs [151]. The conformational change to amyloid sheets is crucial for prion-like propagation in ALS [152]. TDP-43 can form amyloid-like species, which may be mediated by the low-complexity C-terminal domain frequently mutated in ALS or the RRM regions [153,154,155]. Nonaka et al. demonstrated that adding the insoluble protein fractions from human ALS or FTLD-TDP tissue lysates induces self-templating TDP-43 aggregation in cultured neuronal cells [156]. Other studies have also documented the prion-like propagation of TDP-43, using a co-culture system of stable cells overexpressing TDP-43 [157,158,159], while others show the axonal uptake and transport of TDP-43 seeds [160]. Recently, in vivo studies have supported these previous in vitro reports, by demonstrating that ALS- or FTLD-derived TDP-43 forms TDP-43 inclusions in the recipient transgenic murine brain or spinal cord [157,161].

### 3.15. Gliosis

Glial cells such as astrocytes, microglia, and oligodendrocytes crucially maintain multiple neural homeostatic functions, including synaptic function, supply of metabolites and neurotrophic factors to neurons, and repairment of damaged neural tissue [162,163,164]. In ALS, astrocyte proliferation and hypertrophy are observed in the postmortem central nerve tissue [165]. ALS and FTLD-TDP transgenic mouse models expressing the human TDP-43 mutant proteins (A315T, M337V) also display astrogliosis [166,167]. Interestingly, the astrocyte-specific expression of ALS-linked TDP-43 mutant (M337V) in rats recapitulates ALS phenotypes, including progressive paralysis, motor neuron loss, and gliosis [168]. This result contrasts with the study of transgenic mice expressing mutant SOD1 only in astrocytes, in which the mice developed no phenotype [169]. TDP-43 cytoplasmic aggregates are also detected in oligodendrocytes [170,171]. Transgenic mice devoid of TDP-43 only in oligodendrocytes develop progressive neurological phenotypes and lead to early lethality accompanied by a progressive reduction in myelination, suggesting that TDP-43 is indispensable for oligodendrocyte survival and myelination [172]. von Hippel-Lindau protein (VHL) promotes the degradation of fragmented TDP-43 at proteasomes with a ubiquitin ligase, cullin-2 (CUL2), and colocalizes with TDP-43 aggregates in oligodendrocytes in postmortem ALS spinal cords, implying that imbalances in VHL and CUL2 may underlie oligodendrocyte dysfunction in ALS [173].

## 4. TDP-43 Pathology in Neurodegenerative Diseases

Although pathologically ubiquitinated and phosphorylated TDP-43 inclusions are commonly linked to neurodegeneration in ALS and FTLD-TDP [5,6], TDP-43 pathology has also been observed in patients with other neurodegenerative diseases such as Alzheimer’s disease (AD), Parkinson’s disease (PD), dementia with Lewy bodies (DLB), Huntington’s disease, corticobasal degeneration (CBD), progressive supranuclear palsy (PSP), Guam parkinsonism-dementia complex (G-PDC), and Perry disease [15].

### 4.1. ALS/FTLD

Strikingly, nuclear depletion and the cytoplasmic accumulation of TDP-43 inclusions are evident in up to 97% of all ALS cases [7]. These TDP-43 inclusions are ubiquitinated and hyperphosphorylated, recognized as a pathological hallmark in ALS [5,6]. Pathological TDP-43 inclusions show varying morphologies, such as neuronal cytoplasmic inclusions with rounded or skein-like appearance, short or long dystrophic neurites, and rare glial inclusions primarily in oligodendrocytes [6,174]. Interestingly, the composition of TDP-43 inclusions in brain tissues differs from their composition in spinal cord tissues, with an increased representation of TDP-43 CTFs in cortical and hippocampal regions, implying that regionally different pathogenic processes may underlie the development of TDP-43 pathology [175]. The distribution of TDP-43 pathology differentiates ALS-FTLD from ALS without FTLD. TDP-43 pathology in extra-motor regions in the cerebrum is strongly associated with cognitive impairment in ALS [176,177] (Figure 3). Moreover, executive dysfunction is characterized by TDP-43 pathology in the orbitofrontal cortex, ventral anterior cingulate, dorsolateral prefrontal cortex, and medial prefrontal cortex. Language dysfunction is associated with TDP-43 pathology in the inferior frontal gyrus (Broca’s area), transverse temporal area (Heschl’s gyri), middle and inferior temporal gyri, and angular gyrus. Verbal fluency dysfunction correlates with TDP-43 pathology in the prefrontal cortex, inferior frontal gyrus (Broca’s area), ventral anterior cingulate, and the transverse temporal area (Heschl’s gyri). Behavioral impairment is related to TDP-43 pathology in the orbitofrontal cortex, ventral anterior cingulate, and medial prefrontal cortex [176]. Semantic aphasia is associated with TDP-43 pathology in the anterior temporal cortex [178].

TDP-43 inclusions in frontal and anterior temporal lobe regions are detected in a subgroup of FTLD, named FTLD with TDP-43 (FTLD-TDP). FTLD-TDP accounts for ~45% of all FTLD cases and is distinguished from other FTLD subgroups with pathological tau (FTLD-tau), fused in sarcoma (FTLD-FUS), and other proteins [7]. FTLD-TDP is further classified into four subtypes (types A, B, C, and D) based on the distribution and morphology of cytoplasmic or intranuclear TDP-43 pathology and clinical features [179,180]. The FTLD-TDP type A is defined as the presence of many small compact or crescent neuronal cytoplasmic TDP-43 inclusions and short dystrophic neurites primarily in the cortical layer 2, which is associated with non-fluent primary progressive aphasia (PPA) and the behavioral variant FTLD (bvFTLD) (with or without ALS) [3,179]. The FTLD-TDP type B is characterized by diffuse or granular cytoplasmic TDP-43 inclusions with relatively few dystrophic neurites throughout all cortical layers, observed in ALS-FTLD and bvFTLD [3,179]. The FTLD-TDP type C presents few cytoplasmic TDP-43 inclusions but abundant long and tortuous dystrophic neurites primarily in cortical layer 2, observed in semantic-variant PPA or temporal-variant bvFTLD [3,179]. FTLD-TDP type D correlates with mutations in the *VCP* gene and has many lentiform neuronal TDP-43 intranuclear inclusions throughout the cortical layers [3,179]. The genetic analysis shows that mutations in *C9orf72*, *GRN*, *CHMP2b*, and *TARDBP* genes are also associated with FTLD-TDP [181]. Although the subtypes of TDP-43 pathology do not always accord with genetics or clinical profiles, mutations in the *GRN* gene typically correlate with type A pathology. *C9orf72* mutations are generally associated with FTLD-TDP type B [179,182].

Interestingly, TDP-43 pathology can also be detected in the cells related to the clinical profiles of ALS/FTLD other than motor and cognitive dysfunctions. ALS/FTLD patients often show unusual eating behavior [183], sleep [184], and energy metabolisms such as glucose intolerance and hyper lipid metabolism [185,186]. Oxytocin and orexin neurons display TDP-43 inclusions in the hypothalamus of ALS patients, in addition to the loss of oxytocin- and orexin-producing neurons related to abnormal eating behavior and sleep in ALS [187]. ALS subjects reduce early-phase insulin secretion in parallel with motor dysfunction, and the nuclear localization of TDP-43 is lost in the islets of autopsied ALS pancreas [188]. ALS patients also exhibit TDP-43 inclusions in the intramuscular nerve bundles from muscle biopsy even before the clinical diagnosis, suggesting that TDP-43 pathology in the intramuscular nerve bundles may be a novel early diagnostic biomarker [189].

### 4.2. Limbic-Predominant Age-Related TDP-43 Encephalopathy (LATE)

Limbic-predominant age-related TDP-43 encephalopathy (LATE) is clinically associated with an amnestic dementia syndrome that mimics Alzheimer’s-type dementia, and its neuropathological change is defined by a stereotypical TDP-43 proteinopathy in older adults, with or without co-existing hippocampal sclerosis pathology. In LATE, TDP-43 inclusions are predominantly confined to the limbic system, the middle frontal gyrus, and the medial temporal lobe. The genetic variation in five genes, *GRN*, *TMEM106B*, *ABCC9*, *KCNMB2* and *APOE*, is associated with LATE [13].

### 4.3. Other Neurodegenerative Diseases

Facial onset sensory and motor neuronopathy (FOSMN) is a rare neurodegenerative disease of motor and sensory neurons, initially developing paresthesia and numbness in a trigeminal nerve distribution and motor manifestations, with limb and bulbar muscle weakness developing later in the course of the illness [190]. Mutations in several familial ALS genes such as *TARDBP*, *SOD1*, *SQSTM1*, *VCP*, and *CHCHD10* have been reported in relation to FOSMN syndrome, and TDP-43-positive intraneuronal inclusions are identified in the brain, spinal cord, and dorsal root ganglia, suggesting that FOSMN is most likely to be a TDP-43 proteinopathy within the ALS-FTLD spectrum [191,192,193,194]. TDP-43-positive inclusions within neurons and oligodendroglia are identified in the brains of patients with AD and DLB, in which a subset of TDP-43-positive inclusions co-exists with neurofibrillary tangles or Lewy bodies, generally found in AD or DLB, respectively, in the same neurons [195]. TDP-43 pathology can be identified in 19–57% of AD cases [196]. Furthermore, TDP-43-positive intraneuronal inclusions have also been observed in the spinal cord and bulbar nuclei of PD patients [197,198]. In Huntington’s disease, TDP-43 is frequently colocalized with huntingtin in the dystrophic neurites and the cytoplasmic inclusions but not in the intranuclear inclusions [199]. TDP-43 immunohistology has also revealed that glial TDP-43 pathology with the staining of astrocytic plaque-like structures and coiled bodies can be identified in 15.4% of CBD cases [200]. TDP-43-positive inclusions can be observed in 26% of PSP cases, as the disease-vulnerable regions such as the amygdala, hippocampus, entorhinal cortex, medial occipitotemporal gyrus, and dorsolateral frontal lobe, are susceptible to TDP-43 pathology [201,202]. Recently, TDP-43 pathology has also been reported in the spinal cord motor neurons in 58% of CBD and 38% of PSP cases [203]. G-PDC, a neurodegenerative disease of Chamorro residents of Guam clinically characterized by either progressive cognitive impairment with extrapyramidal signs or motor neuron dysfunctions, is associated with cortical TDP-43-positive dystrophic neurites and neuronal and glial inclusions in gray and/or white matter, in addition to cortical neurofibrillary and glial tau pathology [204,205]. Perry’s syndrome is an autosomal dominant and early-onset rapidly progressive disease, showing parkinsonism, hypoventilation, depression, and severe weight loss. It exhibits TDP-43-positive neuronal inclusions, dystrophic neurites, and axonal spheroids that are selective for the extrapyramidal system, and it spares the neocortex, hippocampus, and motor neurons with the distribution being distinct from ALS and FTLD-TDP [14]. The causative gene *DCTN1* encodes the largest subunit of the dynactin complex, which associates with the microtubule-based motor protein dynein and is required for dynein-mediated long-distance retrograde transport [206]. Interestingly, DCTN1 is involved in TDP-43 cytoplasmic-nuclear transport, and the dysregulation of DCTN1-TDP-43 interactions triggers the mislocalization and aggregation of TDP-43 [207]. Considering that TDP-43-positive inclusions are detected in various neurodegenerative disorders other than ALS and FTLD-TDP, further investigation is required to elucidate whether TDP-43 inclusions primarily play a crucial role in triggering these disorders or if they are secondarily induced by the other primary aggregating proteins such as tau and α-synuclein.

## 5. Conclusions

TDP-43 has emerged to function as a pivotal protein for cellular homeostasis. Overall, TDP-43 dysfunction due to various factors, such as imbalance of nucleo-cytoplasmic distribution, dysregulations of RNA metabolism, genetic mutations, aberrant post-translational modifications, aggregation, and gain of cytotoxicity, induces the collapse of cellular homeostasis and leads to TDP-43 proteinopathy in ALS and FTLD-TDP. Although the findings do not conclusively prove whether the aberrant TDP-43 protein is fundamentally involved in the pathophysiology of other neurodegenerative diseases, the expanded neurodegenerative spectrum of TDP-43 proteinopathy, including LATE and Perry’s syndrome with TDP-43-positive inclusions, suggests that TDP-43 may, at least in part, mediate neurodegeneration. Therefore, the revelation of the mechanisms involved in TDP-43 homeostasis and dysfunctions will yield novel therapeutic targets against ALS and FTLD-TDP and multiple neurodegenerative diseases.

## Figures and Tables

**Figure 1 ijms-23-12508-f001:**
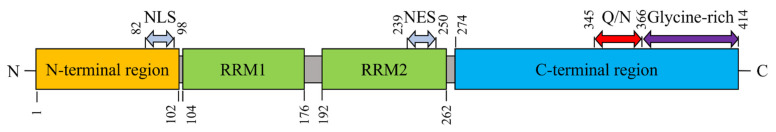
The structure of TAR DNA-binding protein 43 (TDP-43) protein. TDP-43 is composed of an N-terminal region, two RNA recognition motifs (RRM1 and RRM2), a nuclear localization signal (NLS), a nuclear export signal (NES), and a C-terminal region (with a prion-like glutamine-/asparagine-rich [Q/N] region and a glycine-rich region). The numbers represent amino acid positions.

**Figure 2 ijms-23-12508-f002:**
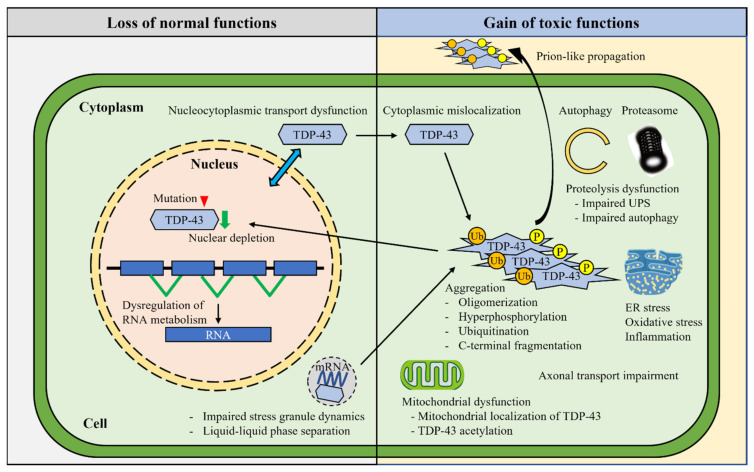
Molecular mechanisms underlying TDP-43 pathogenesis. Various mechanisms have been identified in the TDP-43 pathogenesis. The impairments of TDP-43 autoregulation and nucleocytoplasmic shuttling result in the dysregulation of RNA metabolism and increased cytoplasmic TDP-43 mislocalization. TDP-43 mislocalization triggers various post-translational modifications, including hyperphosphorylation, ubiquitination, acetylation, and C-terminal fragmentation, which facilitate TDP-43 oligomerization and aggregation. The FALS-linked mutations enhance intrinsic aggregation. The nuclear depletion and cytoplasmic aggregation of TDP-43 induce multiple cytotoxic effects, such as aberrant stress granule dynamics, liquid–liquid phase separation, mitochondrial dysfunction, endoplasmic reticulum (ER) stress, impaired axonal transport, and proteolysis dysfunction. TDP-43 aggregates show prion-like cell-to-cell propagation, which may confer the disease progression.

**Figure 3 ijms-23-12508-f003:**
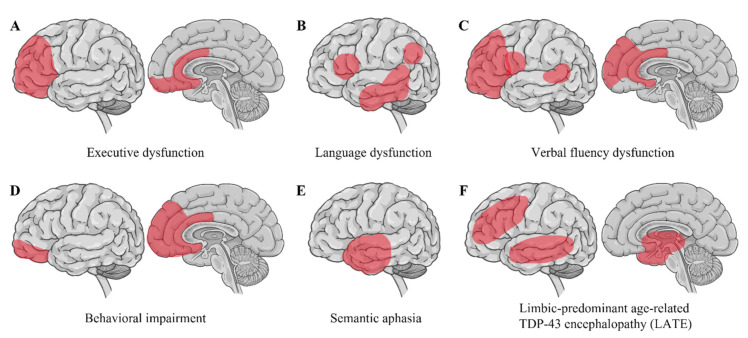
The distributions of TDP-43 pathology in the extra motor regions. The distribution of TDP-43 pathology in ALS-FTLD differs depending on the clinical types of cognitive impairments, such as: (**A**) executive dysfunction; (**B**) language dysfunction; (**C**) verbal fluency dysfunction; (**D**) behavioral dysfunction; and (**E**) semantic aphasia. Furthermore, (**F**) Limbic-predominant age-related TDP-43 encephalopathy (LATE) exhibits TDP-43 pathology in the limbic system, the frontal lobe, and the temporal lobe. The red areas show the distributions of TDP-43 pathology in the cerebrum.

## Data Availability

Not applicable.

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
