# Peer review of "Molecular Dissection of TDP-43 as a Leading Cause of ALS/FTLD"

_ijms, 2022, doi:10.3390/ijms232012508_

Round 1
Reviewer 1 Report
The authors provide an extensive review of the pathogenic mechanisms of TDP-43 in ALS/FTLD, including TDP-43 mutations and post-translational modifications, stress, LLPS, and protein quality control. They also summarize the various neurodegenerative disease with TDP-43 pathology as well as ALS/FTLD. Overall, this review provides a good summary of the extensive information on TDP-43, and is worthy of the readers in this field for reference.
However, there are some points that need to be revised. First, the references cited are relatively old and should be updated. Second, a more detailed overview of the physiological functions of TDP-43 should be provided, especially related to its important functions, such as RNA splicing and nucleocytoplasmic trafficking, which are involved in ALS/FTLD pathogenesis. Finally, the classification of some sections may be misleading and needs to be corrected.
Specific Comments
1. TDP-43 mitochondrial localization depends on mitochondrial internal motifs M1 (aa 35–41), M3 (aa 67 146–150), and M5 (aa 294–300), which are typically composed of a stretch of continuous hydrophobic amino acids [27] (Figure 1).
-- These mitochondria internal motifs are described only in this cited article (Wang, W. et al. The inhibition of TDP-43 mitochondrial localization blocks its neuronal toxicity. Nat. Med. 2016, 22, 869–878). This reviewer cannot find the follow-up studies to validate the role of these domains in TDP-43 function. Are they widely accepted in the field?
2. (Lines 71-74) “The N-terminal region regulates TDP-43 physiological monomer- homodimerization transition and RNA splicing and mediates aggregate formation [29]. This region promotes self-oligomerization of TDP-43 in a concentration-dependent manner and enhances its DNA-binding affinity [30].”
-- More recent references on dimerization/oligomerization of the N-terminal domain should also be taken into account. Recently, dimerization has been implicated mainly in RNA splicing. There are also reports that the N-terminal domain is also involved in LLPS.
Afroz, T. et al. Functional and dynamic polymerization of the ALS-linked protein TDP-43 antagonizes its pathologic aggregation. Nat. Commun. 8, 45 (2017). 
Wang A, Conicella AE, et al. A single N-terminal phosphomimic disrupts TDP-43 polymerization, phase separation, and RNA splicing. EMBO J. 2018 Mar 1;37(5):e97452. doi: 10.15252/embj.201797452. PMID: 29438978.
3. (Lines 79-80) “RRM1 and RRM2 domains in TDP-43 are indispensable for RNA/DNA binding to regulate the transcription, translation, splicing, and stability of mRNA [34].”
-- Authors should describe in more detail the physiological function of TDP-43 in RNA metabolism via RNA binding motifs by citing related references. For example, RNA regulation via splicing of cryptic exons by TDP-43, TDP-43 preferentially binds to UG/TG-rich sequences, and self-regulation via TDP-43 splicing.
Ling, J. P., Pletnikova, O., Troncoso, J. C. & Wong, P. C. TDP-43 repression of nonconserved cryptic exons is compromised in ALS-FTD. Science 349, 650–655 (2015).
Koyama A, Sugai A, et al. Increased cytoplasmic TARDBP mRNA in affected spinal motor neurons in ALS caused by abnormal autoregulation of TDP-43. Nucleic Acids Res. 2016 Jul 8;44(12):5820-36. doi: 10.1093/nar/gkw499. PMID: 27257061.
4. (Line 107) “3.1. Mutant TDP-43 proteins”
In recent years, several reports have been published using knock-in mouse models, which should also be mentioned. This may be discussed in the 3.3.2. Dysregulation of RNA metabolism.
5. (Lines 129-130) “Various post-translational modifications of TDP-43, including cleavage, ubiquitination, phosphorylation, and oligomerization, have been implicated in neurotoxicity in the TDP-43 proteinopathies.”
(Lines 198-211) “3.2.6. Oligomerization”
-- Oligomerization is not a post-translational modification, therefore this session needs to be relocated. For example, it seems reasonable to include this session in "3.4. Gain of toxic functions".
6. (Lines 160-161) “TDP-43 has multiple phosphorylation sites in carboxyl-terminal regions, although the physiological function of TDP-43 phosphorylation remains unknown.”
-- While the physiological function of TDP-43 phosphorylation is controversial, it has been examined in several studies. For example, recent reports suggest that phosphorylation confers a protective effect.
Gruijs da Silva LA, Simonetti F, et al. Disease-linked TDP-43 hyperphosphorylation suppresses TDP-43 condensation and aggregation. EMBO J. 2022 Apr 19;41(8):e108443. doi: 10.15252/embj.2021108443. PMID: 35112738.
7. (Lines 207-211) “The N-terminal region (aa 3–183) of TDP-43 is predicted to be an assembly interface forming an 86-kDa dimeric TDP-43. Although the physiological roles of TDP-43 dimer are unresolved, it is reported that the dimeric TDP-43 induces TDP-43 aggregates in the cultured cells. Moreover, an 86 kDa species is identified in the immunoblot of extracts from the postmortem ALS brains [28]. ”
-- Dimerization and oligomerization of the N-terminal domain is important for physiological functions of TDP-43, such as splicing function, and should be distinguished from "oligomerization as a pre-stage of pathological aggregation". Shiina et al. (28), Fang et al (63), and Afroz et al. may be looking at completely different oligomers.
Afroz T, Hock EM, et al. Functional and dynamic polymerization of the ALS-linked protein TDP-43 antagonizes its pathologic aggregation. Nat Commun. 2017 Jun 29;8(1):45. doi: 10.1038/s41467-017-00062-0. PMID: 28663553.
8. (Line 212) “Loss of normal functions”, (Line 268) “Gain of toxic functions ”
-- Authors describe all TDP-43-linked pathomechanism as “Loss of function” or “Gain of toxic function.” “Loss of function” and “gain of toxic function” are important in ALS pathomechanism, but are not mutually exclusive. For example, stress granules are possibly involved in pathological inclusion body formation, while sequestrating TDP-43 in the cytoplasm may cause loss of function. It seems misleading to categorize all disease mechanisms as “loss of function” or “gain of toxic function”.
9. (Line 233) “3.3.2. Dysregulation of RNA metabolism”
-- With the advancement of new technologies, including RNAsequencing, the link between TDP-43 pathology and RNAs is being extensively uncovered. The authors should discuss and cite more recent reports. The followings are examples.
Mann JR, Gleixner AM, et al. RNA Binding Antagonizes Neurotoxic Phase Transitions of TDP-43. Neuron. 2019 Apr 17;102(2):321-338.e8. doi: 10.1016/j.neuron.2019.01.048. PMID: 30826182.
Hallegger M, Chakrabarti AM, et al. TDP-43 condensation properties specify its RNA-binding and regulatory repertoire. Cell. 2021 Sep 2;184(18):4680-4696.e22. doi: 10.1016/j.cell.2021.07.018. PMID: 34380047.
10. (Lines 241-244) “Mutant TDP-43 (Q331K) reduces splicing processes in target RNAs, such as Kcnip2, Abhd14a, Ctnnd1, and Atp2b1, demonstrating a loss of normal TDP-43 function. In contrast, the Q331K mutant enhances the splicing of other target RNAs, such as Eif4h and Taf1b [70].” 
-- Authors should also discuss the splicing targets of TDP-43, Stathmin-2 and UNC13A, which have recently been reported to be more directly involved in ALS pathogenesis.
Klim JR, Williams LA, et al. ALS-implicated protein TDP-43 sustains levels of STMN2, a mediator of motor neuron growth and repair. Nat Neurosci. 2019 Feb;22(2):167-179. doi: 10.1038/s41593-018-0300-4. PMID: 30643292.
Melamed Z, López-Erauskin J, Baughn MW, Zhang O, Drenner K, Sun Y, Freyermuth F, McMahon MA, Beccari MS, Artates JW, Ohkubo T, Rodriguez M, Lin N, Wu D, Bennett CF, Rigo F, Da Cruz S, Ravits J, Lagier-Tourenne C, Cleveland DW. Premature polyadenylation-mediated loss of stathmin-2 is a hallmark of TDP-43-dependent neurodegeneration. Nat Neurosci. 2019 Feb;22(2):180-190. doi: 10.1038/s41593-018-0293-z. PMID: 30643298.
Brown AL, Wilkins OG, et al. TDP-43 loss and ALS-risk SNPs drive mis-splicing and depletion of UNC13A. Nature. 2022 Mar;603(7899):131-137. doi: 10.1038/s41586-022-04436-3. PMID: 35197628.
Ma XR, Prudencio M, et al. TDP-43 represses cryptic exon inclusion in the FTD-ALS gene UNC13A. Nature. 2022 Mar;603(7899):124-130. doi: 10.1038/s41586-022-04424-7. PMID: 35197626.
--- In addition, this section may be suitable to discuss recently published knock-in mice for TARDBP gene. These works demonstrated the gain of splicing function by missense mutation of TDP-43 in mice.
White MA, Kim E, et al. TDP-43 gains function due to perturbed autoregulation in a Tardbp knock-in mouse model of ALS-FTD. Nat Neurosci. 2018;21(4):552–63.
Fratta P, Sivakumar P, et al. Mice with endogenous TDP-43 mutations exhibit gain of splicing function and characteristics of amyotrophic lateral sclerosis. EMBO J. 2018;37:e98684. This is not an ALS-linked mutation.
Watanabe S, Oiwa K, et al. ALS-linked TDP-43M337V knock-in mice exhibit splicing deregulation without neurodegeneration. Mol Brain. 2020 Jan 20;13(1):8. doi: 10.1186/s13041-020-0550-4.
11. (Lines 244-246) “The examination of splicing patterns of TDP-43 target genes in lower motor neurons of postmortem ALS cases has revealed the widespread dysregulations of mRNA splicing that specifically affected genes involved in ribonucleotide binding [71].” 
-- Authors should also cite the following reference for cryptic exons.
Ling JP, Pletnikova O, Troncoso JC, Wong PC. TDP-43 repression of nonconserved cryptic exons is compromised in ALS-FTD. Science. 2015 Aug 7;349(6248):650-5. doi: 10.1126/science.aab0983. PMID: 26250685.
12. (Lines 257) “3.3.3. Nucleocytoplasmic transport dysfunction”
-- Despite the fact that aberrant nucleocytoplasmic localization of TDP-43 is a pathological hallmark of ALS/FTLD and constitutes an important part of its pathogenesis, the authors have described little about its physiological mechanisms and involvement in the disease. They should refer to the following references to discuss the physiological nucleocytoplasmic transport of TDP-43 and the chaperone effects of the transporter.
Ayala YM, Zago P, et al. Structural determinants of the cellular localization and shuttling of TDP-43. J Cell Sci. 2008, 121:3778-85. doi: 10.1242/jcs.038950. PMID: 18957508.
Nishimura AL, Zupunski V, et al. Nuclear import impairment causes cytoplasmic trans-activation response DNA-binding protein accumulation and is associated with frontotemporal lobar degeneration. Brain. 2010 Jun;133:1763-71. doi: 10.1093/brain/awq111. PMID: 20472655.
Guo L, Kim HJ, et al. Nuclear-Import Receptors Reverse Aberrant Phase Transitions of RNA-Binding Proteins with Prion-like Domains. Cell. 2018 Apr 19;173(3):677-692.e20. doi: 10.1016/j.cell.2018.03.002. PMID: 29677512.
13. (Lines 276-278) “However, failure of the stress response may facilitate the conversion of SGs into pathological inclusions observed in ALS and FTLD [79–82].”
-- Recent studies have reported that pathological inclusions are formed under stress conditions independently of stress granules.
Gasset-Rosa F, Lu S, Yu H, Chen C, Melamed Z, Guo L, Shorter J, Da Cruz S, Cleveland DW. Cytoplasmic TDP-43 De-mixing Independent of Stress Granules Drives Inhibition of Nuclear Import, Loss of Nuclear TDP-43, and Cell Death. Neuron. 2019 Apr 17;102(2):339-357.e7. doi: 10.1016/j.neuron.2019.02.038. PMID: 30853299.
Lu S, Hu J, Arogundade OA, Goginashvili A, Vazquez-Sanchez S, Diedrich JK, Gu J, Blum J, Oung S, Ye Q, Yu H, Ravits J, Liu C, Yates JR 3rd, Cleveland DW. Heat-shock chaperone HSPB1 regulates cytoplasmic TDP-43 phase separation and liquid-to-gel transition. Nat Cell Biol. 2022 Sep;24(9):1378-1393. doi: 10.1038/s41556-022-00988-8. PMID: 36075972.
14. (Lines 269) “Strass granules” should be “Stress granules”.
15. (Lines 295-296) “The low-complexity C-terminal domain of TDP-43 is responsible for 295 the LLPS of SGs and cytoplasmic bodies observed in ALS and FTLD-TDP [84].”
-- Dimerization/oligomerization of the N-terminal domain is also reported to be important for LLPS.
Wang A, Conicella AE, Schmidt HB, Martin EW, Rhoads SN, Reeb AN, Nourse A, Ramirez Montero D, Ryan VH, Rohatgi R, Shewmaker F, Naik MT, Mittag T, Ayala YM, Fawzi NL. A single N-terminal phosphomimic disrupts TDP-43 polymerization, phase separation, and RNA splicing. EMBO J. 2018 Mar 1;37(5):e97452. doi: 10.15252/embj.201797452. PMID: 29438978.
16. (Lines 296-299) “Under physiological conditions, TDP-43 protein is incorporated into SGs, whereas LLPS may convert to cytoplasmic aggregates of TDP-43 in the pathological state, including chronic stresses [87].” 
-- This sentence is misleading and the authors should correct it. TDP-43 is incorporated into stress granules under stress conditions such as oxidative and osmotic stress, not under physiological conditions. Under physiological conditions, it is localized to the nuclear membraneless organelles, such as Gems, PML bodies, or paraspeckles. Furthermore, it is recommended to introduce that Gem formation is impaired in ALS.
Tsuiji H, Iguchi Y, et al. Spliceosome integrity is defective in the motor neuron diseases ALS and SMA. EMBO Mol Med. 2013 Feb;5(2):221-34. doi: 10.1002/emmm.201202303. PMID: 23255347.
Ishihara T, Ariizumi Y, et al. Decreased number of Gemini of coiled bodies and U12 snRNA level in amyotrophic lateral sclerosis. Hum Mol Genet. 2013 Oct 15;22(20):4136-47. doi: 10.1093/hmg/ddt262. PMID: 23740936.
17. (Line 309) “3.4.3. Mitochondrial dysfunction” It is recommended that the following additional recent references be cited regarding mitochondrial dysfunction.
Yu CH, Davidson S, Harapas CR, et al. TDP-43 Triggers Mitochondrial DNA Release via mPTP to Activate cGAS/STING in ALS. Cell. 2020 Oct 29;183(3):636-649.e18. doi: 10.1016/j.cell.2020.09.020. PMID: 33031745.
18. (Line 484-498) “Limbic-predominant age-related TDP-43 encephalopathy (LATE) is clinically associated with an amnestic dementia syndrome that mimics Alzheimer’s-type dementia, and its neuropathological change is defined by a stereotypical TDP-43 proteinopathy in older adults, with or without co-existing hippocampal sclerosis pathology. ..... Mutations in several familial ALS genes such as TARDBP, SOD1, SQSTM1, VCP, and CHCHD10 have been reported in relation to FOSMN syndrome, and TDP-43-positive intraneuronal inclusions are identified in the brain, spinal cord, and dorsal root ganglia, suggesting that FOSMN is most likely to be a TDP-43 proteinopathy within the ALS-FTLD spectrum [155–158].”
-- LATE and FOSMN should be distinguished from ALS/FTLD and these descriptions should be moved to a different section. In particular, LATE is a new disease concept that has attracted much attention in recent years and needs to be described in a new section. FOSMN should be included in "4.2 Other neurodegenerative diseases".
19. (Lines 540-545) “TDP-43 immunohistology has also revealed that glial TDP-43 pathology with the staining of astrocytic plaque like structures and coiled bodies can be identified in 15.4% of CBD cases [175]. TDP-43- positive inclusions can be observed in 26% of PSP cases, as the disease-vulnerable regions such as amygdala, hippocampus, entorhinal cortex, medial occipitotemporal gyrus, and dorsolateral frontal lobe, are susceptible to TDP-43 pathology [176,177]. “
-- Recently, TDP-43 pathology has also been reported in spinal motor neurons in CBD and PSP, therefore, recommending to cite the following as well.
Riku Y, Iwasaki Y, et al. Motor neuron TDP-43 proteinopathy in progressive supranuclear palsy and corticobasal degeneration. Brain. 2022 Aug 27;145(8):2769-2784. doi: 10.1093/brain/awac091. PMID: 35274674.
Reviewer 2 Report
The review manuscript written by Yoshitaka Tamaki and Makoto Urushitani is well written and well cited. I have few minor comments to improve the quality of the manuscript.
First, I think that a paragraph should be added on the role of TDP-43 in regulating the expression of stathmin-2, since it has been discovered as a major player in ALS pathogenesis.
Second, in the section on impairment of protein quality control (ubiquitin and autophagy dysfunction), I would suggest discussing about UBQLN2 and the interaction between UBQLN2 and TDP-43.
Third, to complete the different role of TDP-43, the author may add a few sentences on the role of TDP-43 in inflammation and NF-kB activation.
Fourth, the figure 2 is relevant for the paper but could be improve. The author could for example make a 2 panels figure comparing in more details the loss of normal function and the gain of toxic functions. Each panel can respectively include the mechanisms implicated in both categories and the major other players in each mechanism (already described in the text).
The authors could also consider improving the graphics of figure 1, for example by using more uniform color and using a different shape for the protein domains.
